# Sensorimotor Time Delay Estimation by EMG Signal Processing in People Living with Spinal Cord Injury

**DOI:** 10.3390/s23031132

**Published:** 2023-01-18

**Authors:** Seyed Mohammadreza Shokouhyan, Mathias Blandeau, Laura Wallard, Thierry Marie Guerra, Philippe Pudlo, Dany H. Gagnon, Franck Barbier

**Affiliations:** 1University Polytechnique Hauts-de-France, CNRS, UMR 8201-LAMIH, F-59313 Valenciennes, France; 2Pathokinesiology Laboratory, Center for Interdisciplinary Research in Rehabilitation of Greater Montréal (CRIR), Montréal, QC H3S 1M9, Canada; 3INSA Hauts-de-France, F-59313 Valenciennes, France

**Keywords:** spinal cord injury, physiological time delay, Teager–Kaiser Energy Operator, cepstral analysis, power spectrum, EMG

## Abstract

Neuro mechanical time delay is inevitable in the sensorimotor control of the body due to sensory, transmission, signal processing and muscle activation delays. In essence, time delay reduces stabilization efficiency, leading to system instability (e.g., falls). For this reason, estimation of time delay in patients such as people living with spinal cord injury (SCI) can help therapists and biomechanics to design more appropriate exercise or assistive technologies in the rehabilitation procedure. In this study, we aim to estimate the muscle onset activation in SCI people by four strategies on EMG data. Seven complete SCI individuals participated in this study, and they maintained their stability during seated balance after a mechanical perturbation exerting at the level of the third thoracic vertebra between the scapulas. EMG activity of eight upper limb muscles were recorded during the stability. Two strategies based on the simple filtering (first strategy) approach and TKEO technique (second strategy) in the time domain and two other approaches of cepstral analysis (third strategy) and power spectrum (fourth strategy) in the time–frequency domain were performed in order to estimate the muscle onset. The results demonstrated that the TKEO technique could efficiently remove the electrocardiogram (ECG) and motion artifacts compared with the simple classical filtering approach. However, the first and second strategies failed to find muscle onset in several trials, which shows the weakness of these two strategies. The time–frequency techniques (cepstral analysis and power spectrum) estimated longer activation onset compared with the other two strategies in the time domain, which we associate with lower-frequency movement in the maintaining of sitting stability. In addition, no correlation was found for the muscle activation sequence nor for the estimated delay value, which is most likely caused by motion redundancy and different stabilization strategies in each participant. The estimated time delay can be used in developing a sensory motor control model of the body. It not only can help therapists and biomechanics to understand the underlying mechanisms of body, but also can be useful in developing assistive technologies based on their stability mechanism.

## 1. Introduction

During human sensory motor control, different sensory information is sent to the Central Nervous System (CNS), which processes the data and sends motor commands to various muscles in order to maintain body stability during activities. However, this process is affected by time delays, including the feedback delay due to neural transmission, the motor command delay due to the information process in the CNS [1] and finally an electromechanical time delay due to muscle activation delays [2]. Estimation of these time delays is crucial because higher values of total delay induce stabilization performance degradation leading to system instability [3]. On the other hand, estimating the time delay can help us to understand the underlying mechanisms of sensory motor control of the body. Many studies have shown that time delay changes with exercise [4] and is longer in patients compared with healthy individuals as well as elderly people compared with young individuals [5,6]. Specifically, time delay estimation allows therapists and biomechanics to have better insight regarding designing exercise in addition to assistive development that can be helpful in the rehabilitation process and in improving their performance during daily activities. Various models or simulations have been developed to figure out the sensory motor control mechanism. Using incorrect parameter values in the system can lead to wrong results, interpretation and subsequently wrong rehabilitation decisions or a nonfunctional assistive device. Physiological time delay is thus a crucial parameter in modeling sitting stability.

People living with a complete spinal cord injury (SCI) have numerous issues in stabilizing their body due to a lack of sensory information and joint torques below their injury level. Specifically, any injury in their lumbar level results in damage to their back and intervertebral muscles, which are crucial in stabilizing the inherently unstable spine [7,8]. Thus, after an SCI, patients use their upper limbs and head rather than their muscles in the lumbar level in order to maintain their stability [9]. Therefore, sitting stability will be the first and most important goal of rehabilitation for them [10]. A better understanding of the underlying mechanism of sitting stability can be helpful in employing the best rehabilitation strategy or assistive technologies for SCI people. In addition, several studies performed multiple experimental tests in the presence of perturbation and developed different models to estimate joint torques, kinematic variables that can be effective in identifying the employed stability mechanisms by SCI individuals. Blandeau et al. [11,12] used a time-delayed 2 DOF H2AT model for sitting stability in SCI people such that the head and both arms could slide relative to the trunk rotating at the lumbar level. In this study, the trunk angle and the position of the head and arm center of mass (COM) were estimated by a nonlinear observer tuned using classical optimization techniques based on Linear Matrix Inequalities (LMI). Convergence towards the experimental trajectories is therefore proven using such methodology. In another investigation [13], they designed a nonlinear PI descriptor observer to estimate the body kinematics and unknown inputs in an H2AT model. Guerra et al. [14] estimated the inputs in the H2AT model by an application-oriented control law. This problem resumes in stabilizing an open-loop unstable underactuated nonlinear system with a time-varying delayed control input, which is a difficult problem to control, and was solved efficiently in [14]. Furthermore, in another study [15], a new model was developed for SCI patients to understand the underlying mechanisms of their body sensory motor control system. However, though these studies developed various models to understand the stability mechanisms in SCI patients, these models cannot be used to estimate the time delay value, which is crucial in stabilizing the employed models, and the stability strategy can be changed with different values of time delay.

Other studies tried to estimate the physiological time delay in healthy people and in patients by different approaches of simulation, experimental and combined strategies. The authors of [16] used an experimental protocol and were able to estimate the time delay between 66 to 99 milliseconds in healthy and lower back pain patients by analyzing the electrical muscle activity (EMG) in the presence of an external perturbation in both anterior–posterior and medio-lateral directions in seated balance. Other investigations were also able to estimate the total physiological time delay by analyzing the EMG data for healthy controls and patients [17,18,19] in seated balance and stance balance [4,20,20,21]. Instead of EMG, other studies estimated longer time delays by focusing on COP and kinematic data [22,23]. On the other hand, numerous investigations performed data analysis to estimate the time delay by using multiple clinical data including EMG, center of pressure or joint torque and kinematic data [24,25,26,27,28,29,30,31,32,33].

In addition, several studies [34,35,36,37,38,39,40] developed models to estimate not only the time delay but also other parameters such as joint torques, stiffness, damping, etc. In these studies, a model with multiple unknown parameters was developed in which the parameters were determined using experimental trajectories and an optimization approach. Furthermore, some other investigations used different techniques such as Kalman filter [41], Cepstral analysis in the time–frequency domain [42] and frequency analysis [43] to estimate the time delay. Despite the fact that numerous studies estimate the time delay by different signal processing approaches in healthy and varieties of patients, to the best of our knowledge, no study was conducted for time delay estimation in SCI people during sitting stabilization. Therefore, the main motivation of this study is to estimate the physiological time delay in SCI patients during seated balance through four classical methodologies found in the literature. Regarding the novelty of this work, to the best of our knowledge, we found no study in the literature dealing with the following: 1. the time delay estimation in SCI people during sitting stability and 2. the comparison of various methods for time delay estimation. The article is organized as follows. Section 2 presents the Materials and Methods section with participants’ characteristics, data acquisition and experimental protocol. Results and all estimated values of time delay are shown in the Section 3. In the Section 4, results are discussed and compared with other studies. Finally, the Section 5 presents perspectives and closes the paper with a conclusion.

## 2. Materials and Methods

### 2.1. Participants

Seven complete SCI subjects (ASIA-A, level of injury above T6) with mean age 39.7 years (SD 12.4) participated in this study. Ethical approval was obtained from the Research Ethics Committee of the Center for Interdisciplinary Research in Rehabilitation of Greater Montreal (CRIR-1083-0515R). The participants read and signed the informed consent form prior to initiating the measurements. Physical characteristics of participants are shown in Table 1.

### 2.2. Experimental Protocol

Participants were asked to maintain their sitting stability on a height-adjustable table without back support with hip and knees flexed to 90°, feet resting on the floor and upper limbs flexed to 90° at the elbow level. When sitting stability was achieved, a light destabilizing force was randomly applied at the level of the third thoracic vertebra between the scapulas. The destabilizing force was generated via an impact with a foam-coated wooden pole such that a pressure sensor was added on the tip to define the contact instant (see Figure 1). After one or two familiarization trials with the destabilizing force, each subject completed a minimum of 11 acquisitions. The start time of the trial was vocally announced to participants, at which time they rose their arms and maintained their stability before the perturbation. Their stability was visually evaluated by the examiner, and the time instant was recorded by a synchronized hand switch. Then, the perturbation was exerted at a random time, and participants should have regained their stability. Their status was again visually assessed, and the time instant recorded when they achieved their stability.

The experimental protocol total duration was about one hour. The first half hour was dedicated to welcoming the subject, receiving his/her agreement for participating in the experiment and finally installing the EMG. The acquisition lasted for approximately 20 min with up to 1 min break between each acquisition. The last 10 min were dedicated to instrumentation removing and obtaining feedback from the subject.

### 2.3. Instruments and Data Acquisition

EMG signals were recorded from the following eight upper limb and trunk muscles: Deltoid Anterior (DA), Deltoid Posterior (DP), Pectoralis Major Clavicular (PMC), Pectoralis Major Sternal (PMS), Biceps Brachii (BB), Triceps Brachii (TB), Trapezius Descending (TD) and Latissimus Dorsi (LD). The skin area was cleaned with alcohol wipes and the electrodes were attached in pairs with a center-to-center distance of 25 mm, based upon recommendations reported in the previous literature [44]. After similar skin preparations, a ground electrode was attached to the anterior aspect of the leg over the tibial bone. The EMG signals were recorded with a commercially available EMG system (TeleMyo 900, Noraxon, Scottsdale, Arizona, USA). All EMG signals and hand switch data were sampled at 1200 Hz.

### 2.4. Data Analysis

In this study, two strategies in the time domain (first and second strategies) and two strategies (third and fourth strategies) in the time–frequency domain were used in order to estimate the time delay in SCI patients by analyzing the EMG data. In addition, the time between the earliest and latest muscle onset was computed as the range of muscle onset. All data analyses were performed with Matlab R2022b software.

#### 2.4.1. First Strategy

At first, all EMG signals were analog filtered using a band pass filter between 30 to 500 Hz by 6th order Butterworth filter, rectified and then low-pass filtered at 100 Hz [45]. The mean and standard deviation (SD) of the signal were computed between 1.5 to 0.5 s immediately before the perturbation. Response onset latencies were determined as the time at which the rectified EMG signal exceeded a threshold of 2×SD above the mean baseline for a period of at least 25 data points [4,46,47]. EMG onset latencies were computed for all muscles and then the average and SD were calculated for all trials in all subjects.

#### 2.4.2. Second Strategy (TKEO)

In this strategy, the raw data were first rectified and high pass filtered at 20 Hz by 6th order Butterworth filter to remove motion and electrocardiogram (ECG) artifacts. Then, the nonlinear Teager–Kaiser Energy Operator (TKEO) [48] was employed and the data were filtered again (6th order, zero-phase low-pass filter at 50 Hz) for smoothing the signal. The TKEO function (*T*) is defined as below:(1)$$T\left[x\left(n\right)\right]=x^{2}\left(n\right)-x\left(n+1\right)x\left(n-1\right)$$
where x represents the rectified and filtered EMG signal and n the sample value. The onset of the muscles defined when the mean value of the smoothed signal exceeded a threshold of the mean plus two standard deviations away from the baseline for more than 25 consecutive samples [32,49]. The mean and SD of baseline were computed from 1.5 to 0.5 s right before the perturbation. Eventually, the response latency was defined as the time between the perturbation instant and onset of each muscle. The response latencies were measured for all muscles and then averaged, and the standard deviations were calculated for all trials and participants. For some acquisitions, the threshold of mean ± 2SD of baseline was not reached by the EMG signal, yielding no onset found. Moreover, when the onset was found below 20 ms, the delay was considered as not found because it was inconsistent with the physiological signal.

#### 2.4.3. Third Strategy (Cepstral Analysis)

The feature of neutral delay-differential equations is mainly that the delay of the neutral part can be detected in the cepstrum of the output signal, which motivated one study [42] to estimate the delay of the acceleration feedback term in stick balancing tasks on kinematic data for healthy individuals. Thus, the cepstral analysis was used in this study as the third strategy for time delay estimation in SCI people. At first, the cepstral transformed signal of each EMG signal was obtained from the smoothed signals of the second strategy (TKEO) as shown in equation 2, in which F and $F^{-1}$ represent Fourier transform and $T\left(n\right)$ is the signal time series after performing the TKEO technique. The frequency domain of 0–0.5 s was examined to find the sharp peaks. The instant of the maximum value was defined as the response onset and the response delay was identified as the time between the perturbation instant and response onset for each muscle. Mean and SD of all muscle onsets were then computed for all trials and subjects.
(2)$$C_{p}=F^{-1}\left\{log\left[F\left\{T\left(n\right)\right\}\right]\right\}$$

#### 2.4.4. Fourth Strategy (Power Spectrum)

In this approach, the power spectrum analysis was used to estimate the physiological time delay in SCI patients. The power spectrum of each smoothed signal [43] from the second strategy (TKEO) was extracted over time and frequency as shown in Equation (3), where ${\left|P\left(f\right)\right|}^{2}$ equals the energy density function over frequency. It was observed that most of the signal power is less than 10 Hz, thus the signal power was averaged between 0 to 10 Hz. It was assumed that the instant of power peaks could demonstrate the response onset. Therefore, the time domain of 0 to 0.5 s was investigated to find the instant of the maximum value. Eventually, the physiological time delay was defined as the time between the perturbation instant and when the averaged power signal reaches its maximum value. Mean and SD of the estimated values were then computed for all trials and participants. In addition, in all strategies, the number of estimated muscle onsets higher than 20 ms and less than 500 ms were found as consistent values with physiological time delay.
(3)$$E=\int_{-\infty }^{\infty }{\left|T\left(n\right)\right|}^{2}dt=\int_{-\infty }^{\infty }{\left|P\left(f\right)\right|}^{2}df$$

The algorithm of each strategy is shown in the Figure 2.

### 2.5. Statistical Analysis

For evaluation of the ECG removing artifacts to form the EMG signals, a statistical metric of the Robust Measures of Kurtosis ($KR_{2}$) [50,51,52] was used in this study using the equation below:(4)$$KR_{2}=\frac{F^{-1}\left(0.975\right)-F^{-1}\left(0.025\right)}{F^{-1}\left(0.75\right)-F^{-1}\left(0.25\right)}-2.91$$
where $F^{-1}$ is the inverse cumulative distribution function (quantile function) of the time series data x. Values $F^{-1}$ (0.975) = −$F^{-1}$ (0.025) = 1.96 and $F^{-1}$ (0.75) = −$F^{-1}$ (0.25) = 0.6745 were obtained for the standard Gaussian distribution. Thus, $KR_{2}$ is zero if the data x has Gaussian distribution. The methods to evaluate statistical characteristics in estimating the Probability Density Function (PDF) shapes of EMG signals were composed of two stages. First, the PDF was estimated by kernel smoothing with a Gaussian kernel [53] from all time points, and this smooth density was discretized to 1001 bins of width 0.01 that partitioned the range from −1 to +1. Eventually, the average and standard deviation of $KR_{2}$ were calculated over all trials and subjects. Spearman and Pearson correlation coefficients were calculated to evaluate correlation in muscle activation sequence and delay value in all trials and subjects, respectively [54]. In addition, the hypothesis of distribution in the normal family was examined for the values of all estimated values for 8 muscles and 4 strategies. Significant effects of muscles and strategies (8 × 4) were evaluated by a two-way ANOVA on the dependent variable of estimated time delay. The effect was considered significant if the *p*-value was less than 0.05.

## 3. Results

The EMG signal and power spectrum of one subject are shown in Figure 3 and Figure 4, respectively, and all dashed lines represent the perturbation instant. During the first 4 s, the subject keeps his arms down on his lower limbs, which explains the low EMG activity. Mean and SD values of all estimated time delays based on four strategies are demonstrated in Figure 5. The results show that the third and fourth strategies estimated longer time delays compared with the other strategies in most muscles. In addition, the sequence of muscle activation is shown by numbers in each column bar of mean value. It is clear that the sequence of muscle activation changes based on the employed strategies for the estimation. However, the Trapezius Descending and Deltoid Anterior are activated later than other muscles during the posture stabilization for each strategy.

The results of Kurtosis robustness analysis are shown in Figure 6. It can be observed that the TKEO technique could appropriately remove the ECG and motion artifacts in muscle activities. In contrast, the results for the first strategy showed that the $KR_{2}$ value is far from zero as well for the Gaussian distribution, and its value is even closer to the unfiltered data, which shows less performance in removing ECG and motion artifacts compared with the TKEO strategy.

Descriptive results of four strategies are shown in Table 2. It can be seen that the first and second strategies sometimes failed to find the muscle onset. Furthermore, the result shows that more detections of Latissimus Dorsi onset were found compared with other muscles in the first and second strategies. On the other hand, the third and fourth strategies were able to estimate more time delays consistent with the actual physiological value.

Results of ANOVA test are shown in Table 3. Both muscle and strategy main effects were significant, although their interaction did not show any significant difference.

In the muscle main effect, the estimated time delay of Triceps Brachii was significantly different with Deltoid Anterior, Pectoralis Major Clavicular and Trapezius Descending muscles. Figure 7 presents mean and SD values of all estimated time delays for all muscles in each strategy. The estimated time delay values by each the first and second strategies are significantly different compared to the third and fourth strategies.

## 4. Discussion

As previously specified, the aim of this study was to estimate the muscle onset activation in SCI people by EMG data. To the best of our knowledge, no study has evaluated different EMG signal processing for muscular onset estimation during seated stability of people living with an SCI. According to our analysis, the first and second strategies estimated shorter time delays (mean = 130 ms and 90 ms, respectively) compared with the third and fourth strategies (mean = 230 ms and 220 ms, respectively) in all muscles. It can be interpreted that third and fourth strategies identify the muscle onset in the time–frequency domain and estimate it by using frequency analysis. In addition, the power spectrum showed that the signal power is less than 10 Hz, thus the movement during stability maintenance occurs in low frequency and it takes more time to reach its peak value. Furthermore, the first and second strategies failed to find the onset threshold, showing that the activity of these muscles does not change much compared to the baseline. Hence, these two strategies may not be appropriate in the estimation of time delay in only the time domain during seating stability in patients with an SCI. The EMG signal contains both ECG artifacts and measurement noise. ECG artifacts can affect the first and second strategies more than the others, because any artifacts within the signal frequency bandwidth can increase the amplitude of the measurement and can be mistakenly identified as muscle onset. The measurement noise frequency is much higher than the activation signal frequency, thus the muscle onset can be detected accurately. It seems that the motor control time delay may be identified better in the time–frequency domain compared with the time domain, which is more vulnerable to noises and artifacts. Other studies in the literature also used different multiples of the SD (1, 3 or 4) [17,47,55] to determine the muscle onset, which can change the value of the time delay. In this regard, the third and fourth strategies may be appropriate for time delay estimation with less variability in identifying the muscle onset. However, we found no study estimating the time delay in SCI people, and the results of this study are consistent with the fact that muscle onset happens earlier than torque or body angle response [33]. In addition, the results showed that the estimated time delay in SCI patients is mostly higher compared with healthy individuals [40].

Otherwise, the Kurtosis robustness analysis demonstrated that the TKEO technique could efficiently remove the ECG and motion artifacts from the EMG signal, thus resulting in an accurate muscle onset identification. On the other hand, the results have shown that the first strategy could not remove these artifacts appropriately. Artifacts can then be detected as muscle onset, leading to an erroneous reading of the data, in particular at the level of the command–contraction temporality. In addition, the ANOVA test demonstrated significant differences in each main effect of muscles and strategy on the value of estimated time delay. Each of the first and second groups were significantly different regarding the third and fourth strategies, and the value of estimated time delay in the Triceps Brachii was significantly different compared to the Deltoid Anterior, Pectoralis Major Clavicular and Trapezius Descending muscles. No Spearman correlation coefficient more than 0.5 was found for the sequence of muscle activation in each pair of different strategies. No significative Pearson correlation coefficient was found between the different methods. This can be due to employing different sequence muscle activations for each participant, resulting in different muscle synergies to compensate for the disruption achieved. There was no restriction in the arms motion so that everyone could maintain his/her stability by moving arms in sagittal or axial planes. Thus, it seems rational that no correlation was found due to motion redundancy. The range of EMG onset mean value was highest for the third strategy and lowest for the first strategy.

Several limitations should be mentioned. At first, it should be reminded that only seven persons participated in this study. A high number of repetitive trials were therefore chosen to cope with this small population. Secondly, the perturbation amplitude may change the stabilization strategy employed by the participants; for example, a high amplitude of perturbation can be detected at the cortical level where the time delay is shorter compared with response from CNS. The perturbation amplitude was not normalized in this study as performed in [32,49], and this methodological choice was made in our study to cope with the subjects’ high variability in injury level thus in stabilization performance, which can change the results. Furthermore, participants did not use specific instructions on how to stabilize their body during seated balance, which caused variability in upper limb motion. Last, each participant performed at least 11 trials, which can increase the learning effect. For future works, the value of the estimated time delay will be used in developing models of people living with SCI maintaining their sitting stability. Stability analysis will be studied using a different controller for the CNS. In addition, time delay and other passive elements of their bodies will be estimated by developing a model so that its trajectory is optimized using experimental data, which can help us to estimate more accurate values.

## 5. Conclusions

Two strategies in time domain and two strategies in time–frequency domain were investigated in this study for time delay estimation for people living with an SCI. The TKEO technique efficiently reduced the ECG and motion artifacts compared to the classical filtering approach. However, the first and second strategies failed to find muscle onset in several trials. Time–frequency techniques of cepstral and power spectrum estimated longer time delays due to the lower frequency of motion compared with the two other strategies in the time domain during seated balance. The time–frequency approach appears as a better option when the EMG signal includes artifacts and noises. No Spearman or Pearson correlation coefficient was found in the muscle sequence or delay value in each pair of strategies, which shows each participant used a different strategy and different sequence of muscle activation in maintaining the seated stability. The estimated time delay can help therapists and biomechanics to design more appropriate exercise and develop assistive technologies during or after rehabilitation procedures by better understanding the underlying mechanism of the body sensorimotor control system.

## Figures and Tables

**Figure 1 sensors-23-01132-f001:**
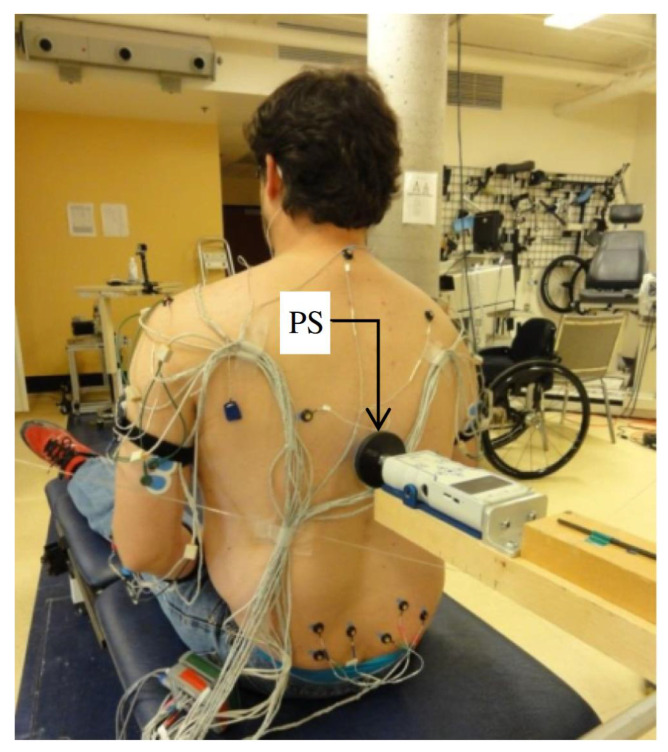
Experimental setup. The perturbation was applied at the level of the third thoracic vertebra.

**Figure 2 sensors-23-01132-f002:**
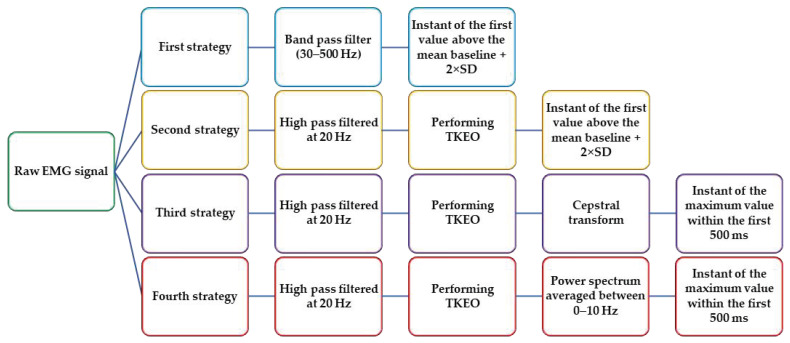
Flow chart of all four strategies used for time delay estimation.

**Figure 3 sensors-23-01132-f003:**
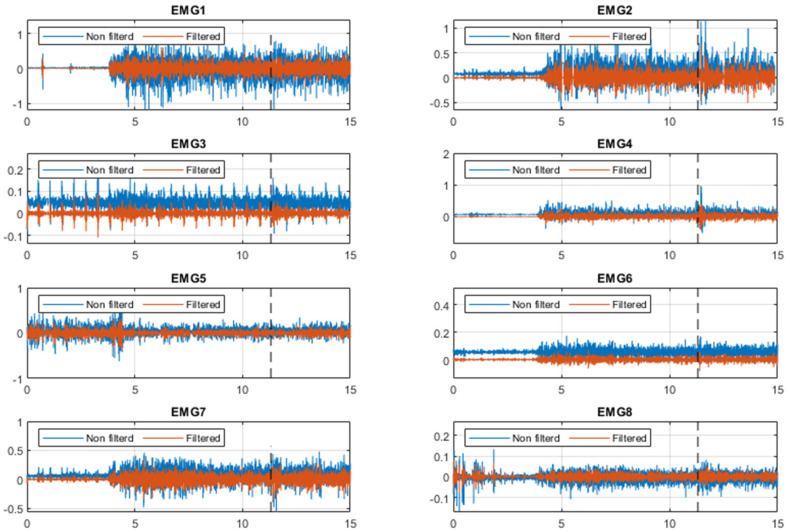
Raw and band pass filtered EMG data for one trial of subject number 7. EMG1 to EMG8 represent Deltoid Anterior, Pectoralis Major Clavicular, Pectoralis Major Sternal, Biceps Brachii, Trapezius Descending, Deltoid Posterior, Latissimus Dorsi and Triceps Brachii, respectively. The black dashed line represents the perturbation instant.

**Figure 4 sensors-23-01132-f004:**
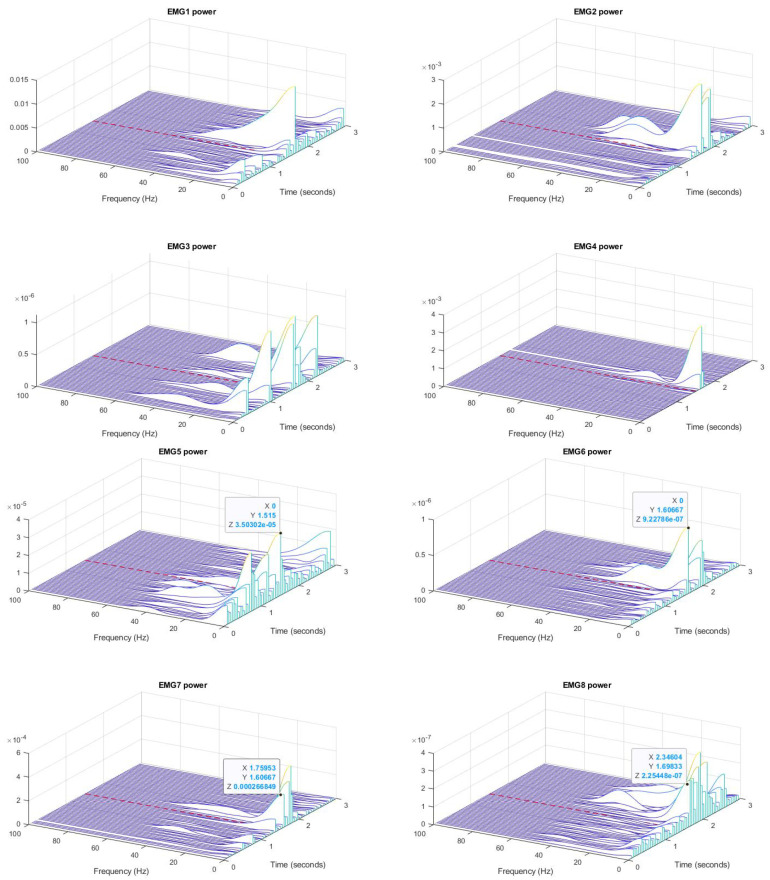
Power spectrum for one trial of subject number 7. EMG1 to EMG8 represent Deltoid Anterior, Pectoralis Major Clavicular, Pectoralis Major Sternal, Biceps Brachii, Trapezius Descending, Deltoid Posterior, Latissimus Dorsi and Triceps Brachii, respectively. The red dashed line represents the perturbation instant.

**Figure 5 sensors-23-01132-f005:**
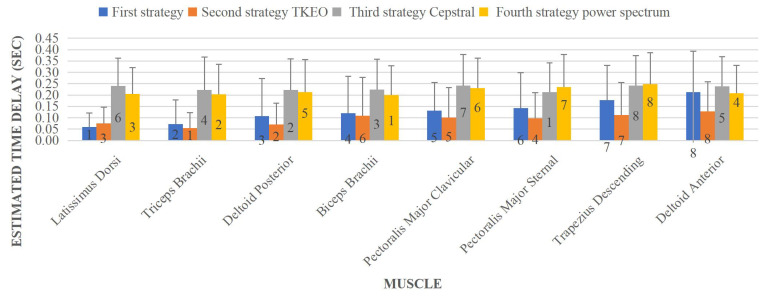
Estimated muscle onset based on different strategies.

**Figure 6 sensors-23-01132-f006:**
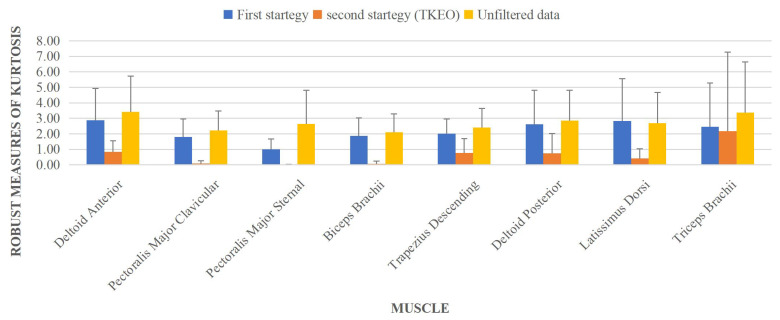
Kurtosis robustness values for both data smoothing techniques and raw data.

**Figure 7 sensors-23-01132-f007:**
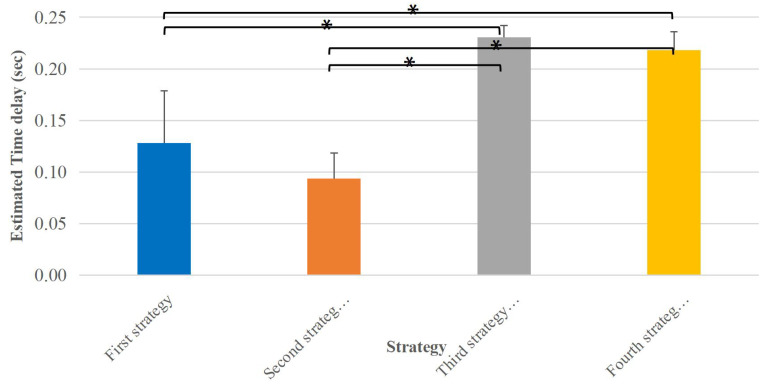
Mean value of all estimated muscle onsets for each strategy. The * stands for significative difference.

**Table 1 sensors-23-01132-t001:** Physical characteristics of participants.

ID	Age	Sex	Weight (kg)	Height (cm)	IMC	Injury Level	ASIA	TIC	Injury Age (Months)
1	33	F	58.5	162.6	22.1	T6	A	0.5	112
2	33	M	59.9	177.8	18.9	T4	A	0	126
3	35	M	76.2	178	24.0	T6	A	0	147
4	31	F	51.7	157.5	20.8	T6	B	0.5	95
5	44	M	63	165	23.1	T6	A	0.5	161
6	57	M	94.8	185	27.7	T4	A	0.5	185
7	45	F	72.1	168	25.5	T4	A	0	131

**Table 2 sensors-23-01132-t002:** Descriptive results for all 4 strategies.

		Muscles	LD *	TB *	DP *	BB *	PMC *	PMS *	TD *	DA *
	Parameters									
*First strategy*	Found value		82	79	77	71	72	79	67	65
	% out of found 82		100	96.3	93.9	86.6	87.8	96.3	81.7	79.3
	Mean value (s)		0.06	0.07	0.11	0.12	0.13	0.14	0.18	0.21
	SD (s)		0.06	0.11	0.16	0.16	0.12	0.16	0.15	0.18
	Consistent value		61	56	63	63	65	64	58	59
	% out of found values		74.4	70.9	81.8	88.7	90.3	81	86.6	90.8
	Range of EMG onset (s)		0.245 ± 0.199							
*Second strategy*	Found value		80	79	80	77	75	79	73	75
	% out of found 82		97.6	96.3	97.6	93.9	91.5	96.3	89	91.5
	Mean value (s)		0.08	0.05	0.07	0.11	0.10	0.1	0.11	0.13
	SD (s)		0.07	0.07	0.09	0.17	0.13	0.11	0.14	0.13
	Consistent value		70	53	63	64	56	64	58	63
	% out of found values		87.5	67.1	78.8	83.1	74.7	81	79.5	84
	Range of EMG onset (s)		0.274 ± 0.222							
*Third strategy*	Found value		82	82	82	82	82	82	82	82
	% out of found 82		100	100	100	100	100	100	100	100
	Mean value (s)		0.24	0.22	0.22	0.22	0.24	0.21	0.24	0.24
	SD (s)		0.12	0.15	0.14	0.13	0.14	0.13	0.13	0.13
	Consistent value		81	80	81	81	77	79	80	80
	% out of found values		98.8	97.6	98.8	98.8	93.9	96.3	97.6	97.6
	Range of EMG onset (s)		0.351 ± 0.084							
*Fourth strategy*	Found value		82	82	82	82	82	82	82	82
	% out of found 82		100	100	100	100	100	100	100	100
	Mean value (s)		0.21	0.2	0.21	0.2	0.23	0.24	0.25	0.21
	SD (s)		0.12	0.13	0.12	0.13	0.13	0.14	0.14	0.12
	Consistent value		78	77	79	79	78	72	75	75
	% out of found values		95.1	93.9	96.3	96.3	95.1	87.8	91.5	91.5
	Range of EMG onset (s)		0.307 ± 0.125							

* LD (Latissimus Dorsi), TB (Triceps Brachii), DP (Deltoid Posterior), BB (Biceps Brachii), PMC (Pectoralis Major Clavicular), PMS (Pectoralis Major Sternal), TD (Trapezius Descending) and DA (Deltoid Anterior).

**Table 3 sensors-23-01132-t003:** ANOVA analysis result.

*Independent Variable*	Estimated Time Delay	
	F-value	*p*-value
*Main Effect*		
*Muscle*	3.79	p<0.05
*Strategy*	62.34	p<0.05
*Interaction*		
*Muscle × Strategy*	1.23	0.21

## Data Availability

The data presented in this study are available on request from the corresponding author.

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
