# Peer review of "Sensorimotor Time Delay Estimation by EMG Signal Processing in People Living with Spinal Cord Injury"

_sensors, 2023, doi:10.3390/s23031132_

Round 1

Reviewer 1 Report

The presented results of the study of rebalancing in patients with Spinal Cord Injury are quite interesting.

However, the article needs significant improvement before further consideration.

I have the following comments.

1. Introduction: specify the purpose of your research. At the beginning of the manuscript reading, the task that you set for yourself is completely incomprehensible. In addition, what is the novelty of your work against the background described in the Introduction.

2. Section "Patients". What are the physical data of patients, at least their weight.

3. Section "Experimental protocol":

3.1. How did you insure the position of patients during the test?

3.2. How did you record the delay time when returning to a steady state? If this was solely based on the experimenter's visual control, then what is the accuracy of such an estimate? It seems that such accuracy is simply not suitable for the analyzed movements of the stabilizing muscles. 

Why was the standard body position sensor not used additionally? 

Such an independent estimate of the stabilization time would be useful for monitoring the results of numerical analysis.

3.3. What is the total duration of the experimental protocol? What are the pauses between mechanical disturbances of stable states?

3.4. Add Figure with fragments of myogram signals recorded during one of the test.

4. Data analysis section:

It is necessary to expand the description of the math methods used.

5. Results section

lines 217-219

You show that different approaches to the analysis of EMG signals demonstrate different scenarios for muscle activation --  "sequences of muscle activation". To what extent can such a result be considered reasonable? Perhaps additional filtering is needed or methods application is not correct?

As a specialist in biomedical signal processing, I believe that an additional criterion for the adequacy of different methods is a certain correspondence between the results of different methods to each other. Here is the reverse situation, which requires explanation.

6. In the Conclusion section, you state that you are using "two strategies in time domain and two strategies in time-frequency domain". This requiresa preliminary explanation in the Methods section, since you do not show signal spectra and do not operate with these concepts.

Author Response

All the co-authors wish to thank the reviewer for the remarks dedicated to improving the quality of this article.

The reviewer’s remarks (in bold) and our answers are provided in details in the PDF file.

Best regards

Reviewer 2 Report

The introduction section must be extended.

The paper structure is absent at the end of the introduction.

The motivation and main contribution must be highlighted.

Revise all explanations of all mathematical variables.

Comparative study with related works must be included.

More recent related works must be included.

Improve the discussion section.

Add more future suggestions.

Improve the abstract and conclusion sections.

Add a high level flow chart and pesudo code of the proposed work.

Author Response

The co-authors wish to thanks the reviewer for all the remarks to improve this article. The remarks are in bold.

The introduction section must be extended.

More description was added as below.

"Furthermore, various models or simulation can be developed to figure out the sensory motor control mechanism which incorrect parameters values of this system can lead to wrong results, interpretation and subsequently wrong rehabilitation decision or nonfunctional assistive device. Physiological time delay is a crucial parameter whom value can result in an unreal model."

The paper structure is absent at the end of the introduction.

The paper structure was added to the introduction as below.

"The article is organized as follows. Section two presents the materials and methods section with participants’ characteristics, data acquisition and experimental protocol. Results and all estimated values of time delay are shown in the third section. In the fourth section, results are discussed and compared with other studies. Finally, the fifth section presents perspectives and closes the paper on conclusion."

The motivation and main contribution must be highlighted.

"Therefore, the main motivation of this study is to estimate the physiological time delay in SCI patients during seated balance through four classical methodologies found in the literature. Regarding the novelty of this work, to the best of our knowledge, we found no study in the literature dealing with 1. the time delay estimation in SCI people during sitting stability and 2. the comparison of various methods for time delay estimation."

Revise all explanations of all mathematical variables.

Mathematical methods were expanded.

Comparative study with related works must be included.

More comparison was added in the discussion section as below.

"However, we found no study estimating the time delay in SCI people, results of this study is consistent with the fact that muscle onset happens earlier than torques or body angle response [33]. In addition, results showed that the estimated time delay in SCI patients is mostly higher compared with healthy individual [40]."

More recent related works must be included.

The cited works have been found through a systematic literature review which is currently under review. This review was performed recently which allowed us to cite very recent works like :

  1. Mohebbi, A.; Amiri, P.; Kearney, R.E. Identification of Human Balance Control Responses to Visual Inputs Using Virtual Reality. J Neurophysiol 2022, 127, 1159–1170, doi:10.1152/jn.00283.2021.
  2. Wang, H.; van den Bogert, A.J. Identification of Postural Controllers in Human Standing Balance. JOURNAL OF BIOMECHANICAL ENGINEERING-TRANSACTIONS OF THE ASME 2021, 143, doi:10.1115/1.4049159.
  3. Nagy, D.J.; Bencsik, L.; Insperger, T. Experimental Estimation of Tactile Reaction Delay during Stick Balancing Using Cepstral Analysis. Mechanical Systems and Signal Processing 2020, 138, doi:10.1016/j.ymssp.2019.106554.

Improve the discussion section.

Discussion has been improved by adding more detail.

Add more future suggestions.

Future works were added as below:

"For future works, the value of the estimated time delay will be used in developing models of people living with SCI maintaining their sitting stability. Stability analysis will be studied using different controller for the CNS. In addition, time delay and other passive elements of their body will be estimated by developing a model so that its trajectory is optimized using experimental data which can help us to estimate more accurate values."

Improve the abstract and conclusion sections.

Abstract and conclusion were improved.

Add a high level flow chart and pesudo code of the proposed work.

The flow chart was added in the manuscript at Figure 2.

Reviewer 3 Report

The paper reports a very interesting study on EMG processing data.  It is a good tool for similar studies regarding biopotential monitoring, very significant from the scientific point of view, and an important contribution to the current state of the art.

My congratulations to the authors.

I only suggest a careful reading of the manuscript, to avoid repeating the same word in consecutive sentences. Example Thus (line 56 and line 57) or Other studies (Lines 80 and 85). Also, some typos can be found: quefrency (line 177) instead of frequency.

Author Response

The co-authors thanks the reviewer for the kind works and remarks.

All repeated words in the manuscript were revised.

Regarding the word “quefrency”, it is correctly used in the context of the cepstral analysis.

Round 2

Reviewer 2 Report

Accepted